# A comparative study of blood cell count in four automated hematology analyzers: An evaluation of the impact of preanalytical factors

Annika Åstrand[1☯]*, Cecilia Wingren[2☯], Claire Walton[3☯], Johan Mattsson[4‡], Komal Agrawal[4‡], Madelene Lindqvist[2☯], Lina Odqvist[2‡], Björn Burmeister[4‡], Steven Eck[5‡], Glen Hughes[6‡], Gabriela Luporini Saraiva[7‡], Anna Schantz[1‡], Ioannis Psallidas[8‡], Christopher McCrae[9☯]

1 Clinical Development, Late-stage Development, Respiratory and Immunology, BioPharmaceuticals R&D, AstraZeneca, Gothenburg, Sweden, 2 Bioscience, Research and Early Development, Respiratory & Immunology, BioPharmaceuticals R&D, AstraZeneca, Gothenburg, Sweden, 3 Biometrics, Late-stage Development, Respiratory and Immunology, BioPharmaceuticals R&D, AstraZeneca, Cambridge, United Kingdom, 4 Translational Science & Experimental Medicine, Research and Early Development, Respiratory & Immunology, BioPharmaceuticals R&D, AstraZeneca, Gothenburg, Sweden, 5 Integrated Bioanalysis, Clinical Pharmacology and Safety Sciences, R&D, AstraZeneca, Gaithersburg, MD, United States of America, 6 Precision Medicine Unit, Biopharmaceuticals R&D, AstraZeneca, Cambridge, United Kingdom, 7 Clinical Development, Late-stage Development, Respiratory and Immunology, BioPharmaceuticals R&D, AstraZeneca, Gaithersburg, MD, United States of America, 8 Clinical Development, Late-stage Development, Respiratory and Immunology, BioPharmaceuticals R&D, AstraZeneca, Cambridge, United Kingdom, 9 Translational Science & Experimental Medicine, Research and Early Development, Respiratory & Immunology, BioPharmaceuticals R&D, AstraZeneca, Gaithersburg, MD, United States of America

☯ These authors contributed equally to this work.
‡ JM, KA, LO, BB, SE, GH, GLS, AS and IP also contributed equally to this work.
* annika.astrand@astrazeneca.com

**Citation:** Åstrand A, Wingren C, Walton C, Mattsson J, Agrawal K, Lindqvist M, et al. (2024) A comparative study of blood cell count in four automated hematology analyzers: An evaluation of the impact of preanalytical factors. PLoS ONE 19(5): e0301845. https://doi.org/10.1371/journal.pone.0301845

**Data Availability Statement:** All relevant data are within the manuscript and its Supporting Information files.

## Abstract

Differential white blood cell counts are frequently used in diagnosis, patient stratification, and treatment selection to optimize therapy responses. Referral laboratories are often used but challenged with use of different hematology platforms, variable blood shipping times and storage conditions, and the different sensitivities of specific cell types. To extend the scientific literature and knowledge on the temporal commutability of blood samples between hematology analyzers, we performed a comparative *ex-vivo* study using four of the most utilized commercial platforms, focusing on the assessment of eosinophils given its importance in asthma management. Whole blood from healthy volunteers with and without atopy (n = 6 +6) and participants with eosinophilic asthma (n = 6) were stored under different conditions (at 4, 20, 30, and 37°C, with or without agitation) and analyzed at different time points (3, 6, 24, 48 and 72h post-sampling) in parallel on the Abbott CELL-DYN Sapphire, Beckman Coulter DxH900, Siemens ADVIA 2120i and Sysmex XN-1000V. In the same blood samples, eosinophil-derived neurotoxin (EDN), eosinophil activation and death markers were analyzed. All platforms gave comparable measurements of cell differentials on fresh blood within the same day of sampling. However, by 24 hours, significant temporal and temperature-dependent differences were observed, most markedly for eosinophils. None of the

**Funding:** AstraZeneca provided financial support through salaries, research materials, and sample costs. AstraZeneca employees were involved in the study design, data collection, analysis, publication decision, and manuscript preparation. AstraZeneca reviewed the publication to ensure medical and scientific accuracy and protect intellectual property. The authors had access to all data in the study and were responsible for submitting the manuscript for publication. The roles of these authors are articulated in the authors' contribution section.

**Competing interests:** The authors, study participants and collaborators have no competing interest in the results. AA, CeW, ClW, JM, KA, ML, LO, BB, SE, GH, GLS, AS, IP, and CMcC are or were employees of AstraZeneca at the time of study conduct and may own stock/stock options.

platforms performed perfectly across all temperatures tested during the 72 hours, showing that handling conditions should be optimized depending on the cell type of interest and the hematology analyzer. Neither disease status (healthy vs. asthma) nor agitation of the sample affected the cell quantification result or EDN release. The eosinophil activation markers measured by flow cytometry increased with time, were influenced by temperature, and were higher in those with asthma *versus* healthy participants. In conclusion, hematology analyzer, time window from sampling until analysis, and temperature conditions must be considered when analyzing blood cell differentials, particularly for eosinophils, via central labs to obtain counts comparable to the values obtained in freshly sampled blood.

## Introduction

*Blood cell differential* is commonly used in clinical practice to identify acute and chronic infections/inflammations and general circulating blood cell status. In the hematology evaluation and optimization of procedures for blood sampling and pre-analytical handling, erythrocytes are often also considered. Standardizing blood storage conditions before analysis of cell differentials has therefore been challenging. The available investigations have often used a single technologic platform at a few sample conditions (commonly refrigeration *versus* room temperature), with different blood cell types in focus. In the early 1990s, it was identified that K2-EDTA was the best anti-coagulant to stabilize blood cells for counting and sizing [1]. In the early 2000s, several reports indicated that the different cell types showed individual stabilities concerning time and temperature [2–4], mostly issues for basophils, eosinophils and neutrophils, which was later confirmed by Pintér et al. [5] when using the CELL-DYN or ADVIA hematology analyzer platforms.

It has previously been suggested that cell differentials should be analyzed within 6 hours of sampling and that refrigeration could stabilize Complete Blood Count (CBC) samples for up to 24–72 hours [2], which was later confirmed as 8 hours at room temperature and up to 24 hours if refrigerated by Pintér et al. [5] using the ADVIA. Imeri et al. [6] were, to our knowledge, the first to demonstrate that assessment differences were partly dependent on the type of hematologic platform technology used, specifically in their head-to-head comparison of platforms from Beckman Coulter, Siemens, and Sysmex. Reassuringly, Meintker et al. [7] demonstrated that the four most common technology platforms (Abbott Sapphire, Siemens Advia 120, Beckman Coulter DxH800, and Sysmex XE-2100) gave comparable results for the white blood cells when analyzed within 4 hours from sampling and stored at room temperature. They also demonstrated a smaller inter-instrument variation compared to manual cell counting with a microscope, which significantly increased the reading precision of the less abundant cells like monocytes and eosinophils ($R^2$ = 0.982–0.995 *versus* 0.647, machines and microscope, respectively). However, none of the above studies compared the four most used technology platforms under pre-analytical conditions such as sampling until analysis time, temperature, and mechanical stress.

Hematology platforms perform differently based on the sample's age when recognizing and counting short-lived cells such as eosinophils and neutrophils [5, 6]. An accurate eosinophil count is for example important in treatment selection of asthma patients. Therefore, we wanted to understand the external factors influencing the divergent results, especially regarding the eosinophils. To close the knowledge gap of a direct head-to-head comparison between commonly used hematology platforms, we investigated the complete white blood cell

differential in blood samples taken from different participants (healthy and with atopic/allergic disease) at different conditions (storage time, temperature, mechanical stress) across four different commercially available analyzers.

## Material and methods

### Included participants

Whole blood was obtained from 12 adult healthy volunteers (5 male; 7 female), 6 with atopy (self-reported previous and/or present asthma and/or allergies), and 6 without. Samples from an additional 6 adults (3 male; 3 female; 18–75 years of age) with physician-diagnosed stable asthma (any of the treatment steps 1–5, GINA 2008), on regular treatment >1 month, with or without inhaled corticosteroids (ICS), and history of eosinophilia (>500 blood eosinophils/μL) were also obtained to capture potential differences in cell count based on cell frequency and activation status. All experiments were performed in accordance with relevant guidelines and regulations (see Declaration section).

### Sample collection and handling

Demographics for the 18 participants in the study are shown in S1 Table and in a schematic of the Schedule of Assessments. A total volume of 95.5 mL of blood was collected per donor and distributed into eight tubes ($K_2$-EDTA, 10 mL lavender/H, #367525 BD Vacutainer) and four EDTA tubes ($K_2$-EDTA, 3 mL lavender/H, #367838 BD Vacutainer) for cell differentials (CBC) at 5 time points: 3h (baseline), 6h, 24h, 48h & 72h, flow cytometry analysis (at 3, 24, 48 & 72h) and plasma collection (all time points), and in one 3.5 mL serum separation tube (Serum separator tubes, SST™ II advance, BD Vacutainer® ref 368498 (Becton Dickinson Sweden AB)) for serum collection (baseline (3h) only). The 3 mL EDTA tubes were transferred at ambient temperature to the Hematology unit, at Sahlgrenska University Hospital (Gothenburg, Sweden), for CBC analysis on the Abbott CELL-DYN Sapphire hematology analyzer. The eight 10 mL EDTA test tubes per donor were transferred at ambient temperature to the Bioscience department at AstraZeneca (Gothenburg) to be analyzed on the Siemens ADVIA 2120i, Beckman Coulter DxH900, and Sysmex XN-1000V hematology analyzers. Two 10 mL EDTA tubes from each donor were placed at four different temperatures: in the refrigerator (at 4˚C), in a box on the bench (room temperature (RT) of 20˚C), and in heating cabinets (at 30 and 37˚C).

**Figure of schedule of assessments.** To mimic the effects of transportation to the analyzing laboratory, mechanical stress was introduced in one of the two tubes at each temperature by shaking them on an IKA® VXR basic Vibrax® orbital shaker (IKA®-Werke GmbH & Co. KG, Staufen im Breisgau, Germany) for four minutes at 400 rpm. The paired tube was left untouched while the shaking was done twice daily, after each CBC measurement and in the afternoon of days 2 and 3, 6 hours after the measurements.

For the biomarker analysis, 300 μL blood from the EDTA tubes were transferred to a 96-well plate (#651201, Microplate, 96 well, PP, V-bottom, Greiner Bio-One International GmbH) after each time point of CBC analysis and centrifuged at 1500 G at 4˚C for 10 min. 20 μL of plasma was transferred to a 96-well plate (#651201, Greiner Bio-One International GmbH) and kept at -70˚C until analyzed with regards to EDN content. Serum was collected in a separate tube at baseline by coagulation for 30 min in RT, followed by centrifugation at 1300×g at RT for 10 min and kept at -70˚C until analysis.

The complete study was run in 3 experiments with n = 6 donor samples per occasion. Experiments 1 and 2 were run on 3+3 healthy participants with and without (w/wo) atopy,

and the third experiment was run on blood from 6 participants with eosinophilic asthma recruited to the Krefting Research Centre (Gothenburg, Sweden).

## Cell differentials using four different hematology analyzers

We used the Abbott CELL-DYN Sapphire, Beckman Coulter DxH900, Siemens ADVIA 2120i, and the Sysmex XN-1000V hematology analyzers to count and compare cell differentials in whole blood. All samples (shaken/not shaken and stored at different temperatures) were measured simultaneously (within 10% time deviation) in the ADVIA, Beckman, and SYSMEX analyzers at AstraZeneca according to the manufacturers' instructions. Blood from 12 donors (n = 6 with asthma and n = 6 healthy (n = 3 with/without atopy)) was analyzed using the Abbott CELL-DYN Sapphire at the Sahlgrenska University Hospital (non-shaken samples only, same temperatures and time windows). An overview of the technical details of each hematology analyzer is given in the supplementary description (reference to MAPSS Optical Cell Detection Technology (hematologyacademy.com), ADVIA® 2120i Hematology System (https://www.medicodistributors.in/siemens-advia-2120i), Evaluation of the New Beckmann Coulter Analyzer DxH 900 Compared to Sysmex XN20 [8], Performance evaluation of the new hematology analyzer Sysmex XN-series [9], and Spurious white blood cell count from a new automated Sysmex XN hematology analyzer [10]).

## Flow cytometry

200 μL of blood/sample was used for analysis by flow cytometry. According to the manufacturer's instructions, the red blood cells were lysed using BD FACS lysing solution (#349202, BD Biosciences). The sample was spun down (300g, 5 min) and washed with PBS. Cells were stained using the LIVE/DEAD™ Fixable Aqua Dead Cell Stain Kit (#L34957 ThermoFisher) following the manufacturers' instructions, spun down (300×g, 5 min), and washed with staining buffer (PBS supplemented with 2% heat inactivated FBS and 1 mM EDTA). Next, samples were incubated with Fc block (#553142, BD Biosciences) and labeled with the following flow cytometry antibodies: CD11b-PE (#555388) CD123-PeCy7 (#560826), CD62L-APC (#559772), HLA-DR-BV786 (#564041), CD14-BV605 (#564054), CD16-alexa700 (#560713) from BD Biosciences and FcεR1-FITC (#334608), Siglec-8-PECF594 (#347109) and CD66b-PerCpCy5.5 (#305108) from Biolegend (modified from Hernandez et al. 2020 [11]). Cells were incubated for 45 min at 4°C, spun down (300×g, 5 min), and washed twice with stain buffer. CountBright Absolute Counting Beads (#C36950, ThermoFisher) were used for determining total cell numbers according to the manufacturer's instructions. The flow cytometry panel is detailed in S2 Table. Samples from 3 different temperatures (4, 20 & 30°C) at 4-time points (3, 24, 48 & 72h) were acquired on a five laser BD LSRFortessa. Gating strategies are shown in S13 Fig.

## Eosinophil derived neurotoxin analysis

Eosinophil-derived neurotoxin (EDN) in plasma and serum was measured with a commercial EDN ELISA (IDK® EDN ELISA, #6811, Immundiagnostik AG, Germany) according to the supplier's instructions with an extended range of calibration curve standards. The microtiter plate, coated with capture anti-EDN monoclonal antibody (#6811, Immundiagnostik AG), was washed five times with 250 μL of wash buffer (#K0001.C.100, Immundiagnostik AG) in each well. After the final washing step, the residual buffer was removed by tapping the plate onto absorbent paper. Lyophilized calibration curve standards and controls (#6811, Immundiagnostik AG) were reconstituted in ultrapure water. 100 μL of the calibration curve standards (0.25–32 ng/mL), controls (C1:[0.7 ng/mL] and C2:[2.94 ng/mL]) and samples diluted 20-fold in assay buffer (#6811, Immundiagnostik AG) were added to the plate wells. Standards

were run in duplicates whereas samples and controls were run in singlicates. The plate was covered and incubated for 1 hour at room temperature on a plate shaker. The contents of each well were then discarded, and the plate was washed. Residual buffer was removed, and 100 μL of the detection anti-EDN peroxidase-conjugated rabbit polyclonal antibody (#6811, Immundiagnostik AG) was added to each well and incubated for 1 hour at room temperature on a plate shaker. After another washing step, 100 μL substrate (tetramethylbenzidine, #K0002.15, Immundiagnostik AG) was added to each well, and the plate was incubated for 15 to 20 minutes, at room temperature, in the dark. 100 μL acidic stop solution (#K0003.15, Immundiagnostik AG) was added, and the plate was read on a SpectraMAX 190 reader (Molecular Devices, Sunnyvale, CA, USA) at 450 nm against 650 nm as a reference. The EDN concentration of unknown samples was calculated by interpolating the standard curve via a 4-parameter logistic fit using SoftMax Pro v5.3 software (Molecular Devices, Sunnyvale, CA, USA).

## Statistical analysis

Linear mixed models were fitted (using SAS version 9.4 software) to each cell type (platelets, neutrophils, lymphocytes, eosinophils, monocytes, and basophils). The participants were nested by their health condition (healthy, allergy/atopics, eosinophilic asthma), and the measurements were repeated (analyzer, time, temperature, shaking) within participants.

The fixed main effects were time window, temperature, analyzer, health condition, and shaking. The 2-way interactions of time, temperature, health condition, shaking, and baseline by analyzer were investigated. Fixed effects were removed in order of lowest F-test until only statistically significant ($p < 0.05$) effects remained. The 3-way interaction between time, temperature, and the analyzer was added if all lower-order interactions were included. The least-squares means (LS-means, predicted population margins) and differences for the 3-way interactions between time, temperature, and instrument were estimated from a model where the residual variance was grouped by analyzer. The interaction between 3-hour cell counts and analyzers was assessed in a second model with 3-hour cell count as a covariate predicting cell counts at later time points.

For the flow cytometry results (live-dead aqua, CD62L, CD66b, CD123, CD11b, FcεR1, and biomarker analysis (EDN), the relationships between time and temperature were modeled using linear mixed models with a random intercept for participants.

Spearman's correlation for EDN in plasma vs serum was done using GraphPad Prism (version 8.4.3).

## Ethical approval and informed consent

Written informed consent was obtained from all participants, and the study was conducted following the 1964 Declaration of Helsinki. Healthy donor blood was obtained through AstraZeneca's Gothenburg Blood donor program (EPN Dnr:2021–04076 including amendment approved 2021-08-24) between December 15th 2021 and January 20th 2022. This ethical approval also contains the Informed Consent Form (ICF) shared and signed by participants. Blood from eosinophilic asthma patients was obtained from the Krefting Research Centre in Gothenburg (EPN Dnr 228–14, Immune Mechanisms in subgroups of asthma including the corresponding ICF D6256M00056) on February 7th until 9th 2022. The experimental procedures were approved by the Central Ethical Review Board of Gothenburg.

## Results

The demographics for the 18 participants in the study are shown in S1 Table. Ten women and eight men were equally distributed over the groups. The ages of the healthy participants and

those with atopy were not reported. Amongst the 6 patients with eosinophilic asthma, the ages ranged from 43 to 70 years. Of these patients, 4 were on ICS.

## Baseline distribution of cell counts

Unadjusted baseline distributions of cell counts by analyzer are shown in Fig 1. The S3 Table shows the differences between analyzers 3 hours post-sampling at room temperature. Of the samples analyzed by the Abbott CELL-DYN Sapphire, a higher proportion were from patients with asthma compared with other analyzers. After adjusting for health conditions due to this imbalance, there were no statistically significant differences in means between hematology platforms when measuring neutrophils, eosinophils, and monocytes. However, baseline platelets, lymphocytes, and basophil measurements had minor differences between analyzers; Abbott CELL-DYN Sapphire and Sysmex XN-1000V had higher counts for lymphocytes and platelets, and the Beckman Coulter DxH900 analyzer had higher counts for basophils.

### 3h Cell differentials at Baseline with the 4 different platforms

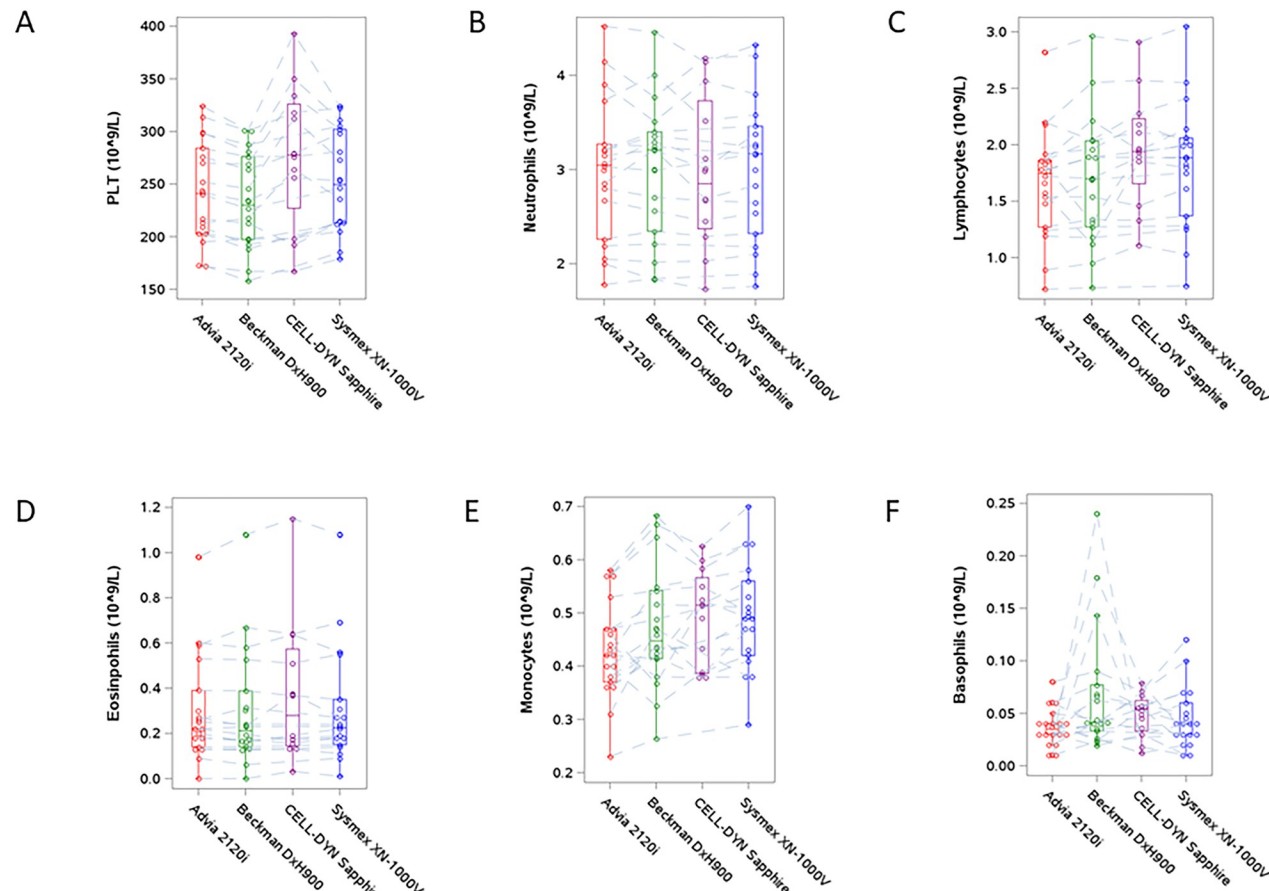

**Fig 1. Cell differential description for the four hematology analyzers (median with interquartile (first to third quartile) range (IQR) with whiskers up to largest observation that falls within 1.5 times the IQR and down to smallest observation falling within this distance.** The observations outside the whiskers boundary are plotted as outliers) A. Platelet count (PLT), B. Neutrophils, C. Lymphocytes, D. Eosinophils, E. Monocytes, and F. Basophils. The platelet count was significantly higher in the CELL-DYN Sapphire platform than in Siemens ADVIA 2120i and Beckman Coulter DxH900 (p = 0.027 and p = 0.002, respectively). Lymphocytes were significantly higher in the Sapphire and XN-1000V platforms vs. the 2120i (p = 0.006 and p = 0.024, respectively), and basophils were significantly lower in the XN-1000V platform vs. DxH900 (p = 0.038). Otherwise, there were no other statistically significant differences between 3-hour baseline cell counts among the 4 hematology analyzers tested.

## Comparison between baseline (3h) and 6 hour cell counts

All analyzers also gave similar results at 6 hours post-sampling at all temperatures as compared to the 3-hour baseline value (S4A–S4F Table and S1 Fig). Correlations (Spearman's rank, r) between the counts at the two time points on all analyzers were between 0.88 and 0.98 for platelets, neutrophils, lymphocytes, and eosinophils (S1A–S1D Fig), except for platelets measured by XN-1000V where it was 0.71. Correlations were lower for monocytes and basophils (S1E–S1F Fig), particularly for the DxH900 (r = 0.72 and 0.31, respectively).

## Analysis of total white blood cell counts

Spaghetti plots (one line per donor) for total white blood cells (WBC) by all analyzers, conditions, temperatures, and time are shown in S2 Fig. Generally, total WBC is more stable in blood samples stored at 20˚C (all timepoints) or 4˚C within 24 hours.

## Analysis of cell differential counts

There were no differences in mean levels between shaken and non-shaken samples for any cell type; therefore, these were considered and treated as duplicate values per participant at each temperature and timepoint. The cell differential counts by time are shown as LS-means (95% CI) from the linear mixed models in Figs 2–4 for 4˚C, 20˚C, and 30˚C, respectively. The baseline mean reference line on each figure was calculated from the 3-hour results per cell type measured by the DxH900, 2120i, and XN-1000V analyzers (Sapphire results were excluded due to the imbalance mentioned above). Most cells were dramatically affected at 37˚C by 24 hours, LS means are shown in S3–S8 Figs for platelets, neutrophils, lymphocytes, eosinophils, monocytes, and basophils, respectively, and differences between means at baseline (3h, 20˚C) and later timepoints and temperatures shown in S4A–S4F Table. A positive difference (3hrs measurement minus later timepoint measurement greater than zero) represents an underestimation whereas a negative difference (3hrs measurement is less than later timepoint measurements) represents an overestimation.

## Cell counts in refrigerated blood samples

Cell counts for samples kept at 4˚C (Fig 2) and beyond 24 hours from blood sampling showed that DxH900, XN-1000V and 2120i were underestimating neutrophils (S4B Table, difference in LS means in cells x$10^9$/L (95% confidence interval)); DxH900 at 3hrs minus 48hrs was 0.931 (95% CI 0.693, 1.169), 72hrs 1.363 (1.125, 1.600)), XN-1000V (48hrs 0.809 (0.535, 1.083), 72hrs 1.294 (1.019, 1.570)) and 2120i (48hrs 0.759 (0.568, 0.949), 72hrs 1.126 (0.935, 1.316)). Scattergrams from the 2120i analysis of refrigerated samples showed infiltration of neutrophils in the eosinophil gate (S9 Fig). DxH900 was overestimating lymphocytes (S4C Table, 48hrs -0.288 (-0.447, -0.129), 72hrs -0.263 (-0.422, -0.103)) and monocytes (S4E Table, 48hrs -0.108 (-0.173, -0.043), 72hrs -0.086 (-0.151, -0.022)) and 2120i was underestimating lymphocytes (S4C Table, 48hrs 0.213 (0.152, 0.275), 72hrs 0.245 (0.184, 0.307)) and overestimating basophils (S4F Table, 48hrs -0.109 (-0.212, -0.007), 72hrs -0.125 (-0.288, -0.023)). By 72 hours, analyzers underestimated platelets (S4A Table, 2120i 16.722 (8.258, 25.287), DxH900 14.150 (6.119, 22.181), and XN-1000V 18.389 (-0.235, 37.013, p = 0.053)). Abbott CELL-DYN Sapphire showed no difference of statistical significance from the 3-hour baseline for any cell type for samples kept at 4˚C up to 48 hours post-sampling (Fig 2, S4A–S4F Table).

## Cell counts in blood samples kept at room temperature

Cell counts (difference in LS means in cells x$10^9$/L (95% confidence interval)) for samples kept at 20˚C (Fig 3) and past 24 hours showed that neutrophils were overestimated by DxH900 at

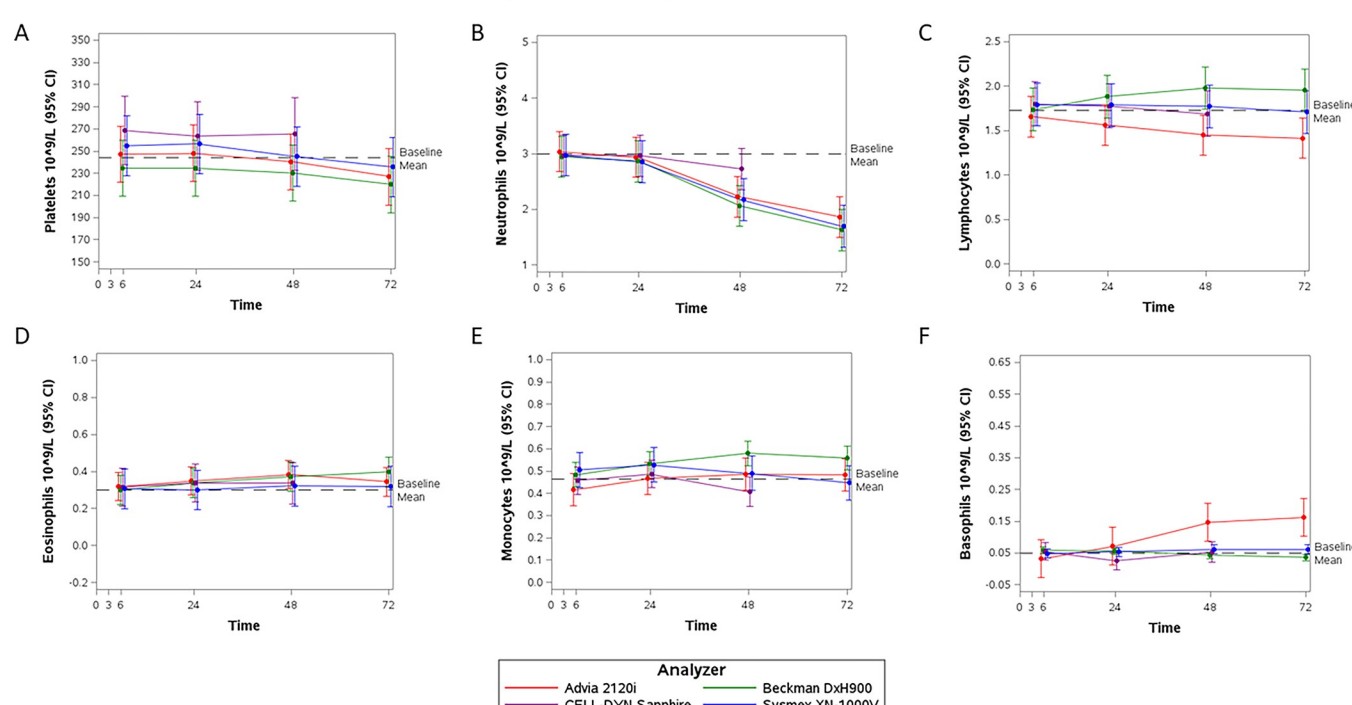

**Fig 2. Ability of the four hematology analyzers to give stable cell differential assessments up to 72 hours in whole blood samples kept at 4˚C (LS means ±95%CI per platform.** Baseline Mean is the average of the 2120i, DxH900, and XN-1000V platforms, n = 18). Results of the linear mixed models (differences in LS means versus 3-hour 20˚C baseline) for each cell by time point and temperature are presented in S4 Table. A. Platelets, B. Neutrophils, C. Lymphocytes, D. Eosinophils, E. Monocytes, and F. Basophils.

48hrs (-0.337 (-0.574, -0.099) and 72hrs -0.644 (-0.880, -0.405)), Sapphire at 48hrs (-0.340 (-0.541, -0.140)) and by 72 hours also by XN-1000V (72hrs -0.402 (-0.676, -0.128), S4B Table). Beyond 24 hours, lymphocyte results were lower than at 3 hours in the 2120i (48hrs 0.182 (0.121, 0.244), 72hrs 0.276 (0.215, 0.337), S4C Table). Eosinophil counts were lower *versus* the 3-hour baseline values in the 2120i (24hrs 0.086 (0.043, 0.129, 48hrs 0.151 (0.108, 0.194), 72hrs 0.153 (0.110, 0.195)) and Sapphire (24hrs 0.156 (0.051, 0.261) and 48hrs 0.285 (0.171, 0.399)) whereas eosinophil counts remained constant over all time points for DxH900 and XN-1000V (S4D Table). Monocyte counts were lower after 24 hours in the DxH900 (48hrs 0.171 (0.106, 0.235), 72hrs 0.287 (0.222, 0.351)) and XN-1000V (72hrs 0.259 (0.142, 0.376)) (S4E Table). At this temperature, basophil counts remained constant over time for all analyzers (S4F Table). However, due to the higher baseline basophil counts for the DxH900, values were lower at the later time points (6hrs 0.029 (0.014, 0.044), 24hrs 0.022 (0.007, 0.037), 48hrs 0.035 (0.020, 0.050), 72hrs 0.037 (0.022, 0.052)).

## Cell counts in blood samples kept at 30˚C

Cell counts (difference in LS means in cells $x10^9$/L (95% confidence interval)) for samples kept at 30˚C (Fig 4) and beyond 6 hours showed that the Sapphire underestimated platelets at 24hrs by 53.250 (26.046, 80.454) and at 48hrs by 54.241 (24.851, 83.639), S4A Table. Underestimation of eosinophils was observed in the 2120i (24hrs 0.125 (0.083, 0.168), 48hrs 0.136 (0.093, 0.179) and 72hrs 0.089 (0.046, 0.132)) and Sapphire (24hrs 0.181 (0.075, 0.286) and 48hrs 0.306 (0.193, 0.420)), S4D Table, and 2120i was overestimating monocytes (24hrs -0.120

LS means from models with time*temperature*analyzer interaction at 20°C

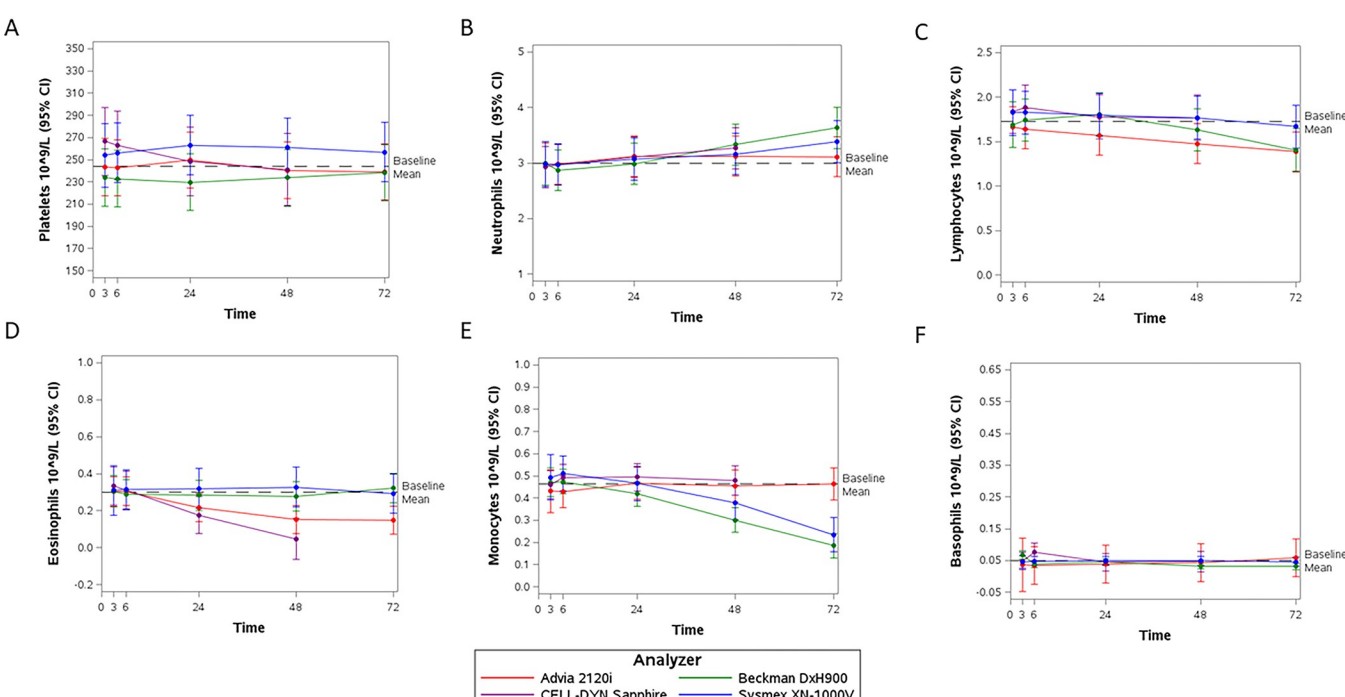

**Fig 3. Ability of the 4 hematology analyzers to give stable cell differential assessments up to 72 hours in whole blood samples kept at 20°C (LS means ±95%CI per platform.** Baseline Mean is the average of the 2120i, DxH900, and XN-1000V platforms, n = 18). Results of the linear mixed models (differences in LS means versus 3-hour 20°C baseline) for each cell by time point and temperature are presented in S4 Table. A. Platelets, B. Neutrophils, C. Lymphocytes, D. Eosinophils, E. Monocytes, and F. Basophils.

(-0.227, -0.012) and 48hrs -0.110 (-0.218, -0.003)), S4E Table. Overestimation of neutrophils was observed by DxH900 (48hrs -0.479 (-0.718, -0.240) and 72hrs -0.644, (-0.882, -0.407)), Sapphire (48hrs -0.279 (-0.480, -0.079)) and XN-1000V (48hrs -0.362 (-0.637, -0.087) and 72hrs -0.544 (-0.818, -0.270)), S4B Table, and underestimation of lymphocytes (DxH900 48hrs 0.323 (0.163, 0.483) and 72hrs 0.633 (0.474, 0.792)), XN-1000V 48hrs 0.374 (0.206, 0.541) and 72hrs 0.821 (0.654, 0.988)), S4C Table, and monocytes (DxH900 24hrs 0.142 (0.077, 0.207), 48hrs 0.317 (0.252, 0.382) and 72hrs 0.315 (0.250, 0.379), XN-1000V 48hrs 0.281 (0.164, 0.399) and 72hrs 0.135 (0.019, 0.252)), Sapphire (48hrs 0.238 (0.166, 0.311)), S4E Table, 2120i was overestimating basophils (72hrs -0.520 (-0.622, -0.417)), S4F Table.

Differential cell count for blood kept at 37°C is shown per individual cell type in supplementary material only (S3–S6 Figs, S10 and S11 Figs, S4A–S4F Table) as most measurements were significantly different from baseline.

## Impact of mechanical stress on cell counts

There were no differences in means or variance estimates between shaken and non-shaken sample results except for an unexpectedly high variance for platelet counts in the non-shaken samples. On further investigation, this was due to one sample measuring high on DxH900 and XN-1000V at 24 hours and 4°C, non-shaken, which was removed from further analysis.

## Variance within analyzers

The variance estimates from the linear mixed models are shown in S5 Table. The most significant variance was associated with the XN-1000V results for all cell types except platelets, where

LS means from models with time*temperature*analyzer interaction at 30°C

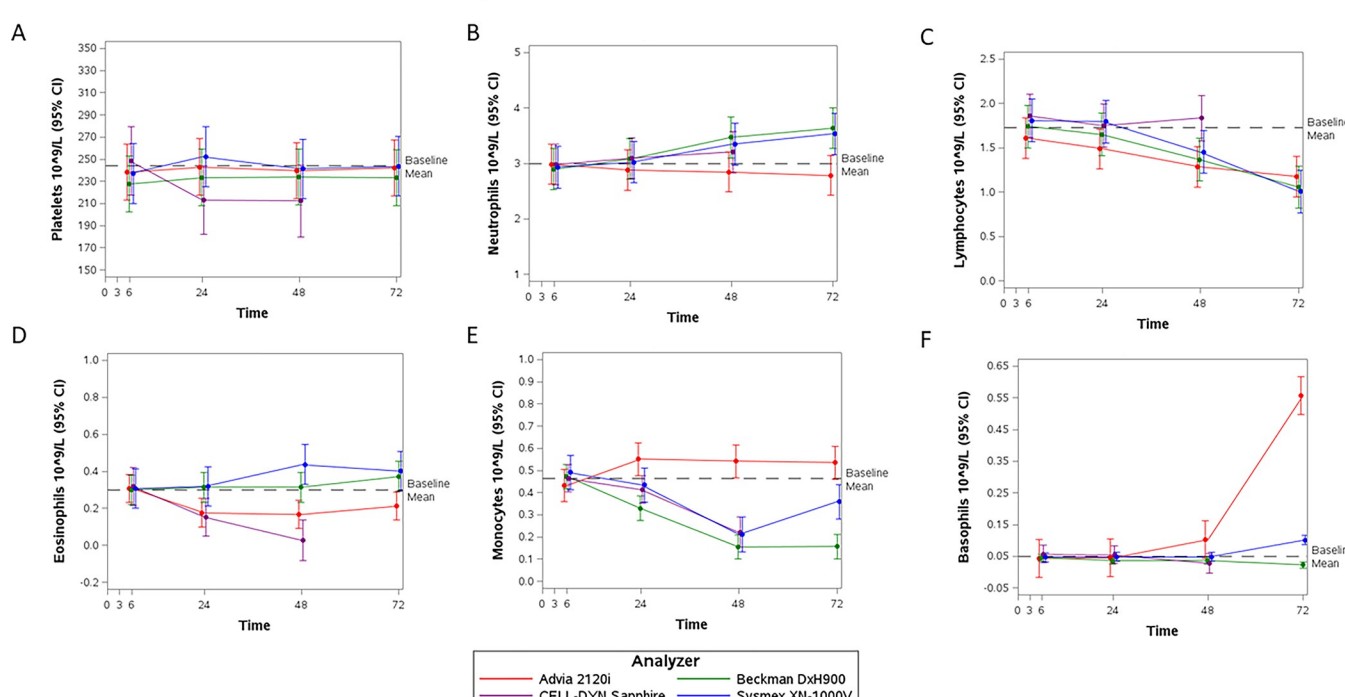

**Fig 4. Ability of the 4 hematology analyzers to give stable cell differential assessments up to 72 hours in whole blood samples kept at 30˚C (LS means ±95%CI per platform.** Baseline Mean is the average of the 2120i, DxH900 and XN-1000V platforms, n = 18). Results of the linear mixed models (differences in LS means *versus* 3-hour 20˚C baseline) for each cell by time point and temperature are presented in S4 Table. A. Platelets, B. Neutrophils, C. Lymphocytes, D. Eosinophils, E. Monocytes, and F. Basophils.

its variance was similar to that observed for Sapphire. For basophils, the analyzer with the greatest variance in results was the 2120i.

## Eosinophil counts

Eosinophils were the cell type that differed the most between the analyzers. Fig 5 shows the unadjusted means for eosinophils by time, temperature, analyzer, and health condition. Spaghetti plots for individual participants per health conditions are shown in S6 Fig.

S6 Table shows the summary statistics for eosinophils by analyzer at 3 hours and 20˚C (baseline). On average and at baseline, healthy participants without atopy had around 0.106 to 0.142 x10$^9$ cells/L depending on the analyzer, while we observed 0.190 to 0.232 x10$^9$ cells/L in healthy participants with atopy. The means of patients with eosinophilic asthma ranged from 0.552 to 0.579 x10$^9$ cells/L depending on the analyzer.

The observed drop from baseline in eosinophil cell count with the 2120i and Sapphire were in absolute numbers larger in those with eosinophilic asthma *versus* healthy participants (see S7 Fig, S7 Table). However, as a percentage change from baseline (S8 Table), the drop was similar regardless of the health condition in the 2120i, with a loss of approximately 50% compared to the 3-hour eosinophil count, and in the Sapphire, a loss of over 75% of cells for the majority of samples (7 out of 9) by 48 hours at 20˚C.

The interaction between baseline eosinophils and analyzer is illustrated in S8 Fig. The DxH900 and XN-1000V analyzers' predictions were close or parallel to the x = y reference dashed line. In contrast, the 2120i's and Sapphire's predicted values were lower than expected in samples with higher baseline levels. At 6 hours for all temperatures and at 4˚C for all times

## Eosinophils by time, temperature, analyzer and condition

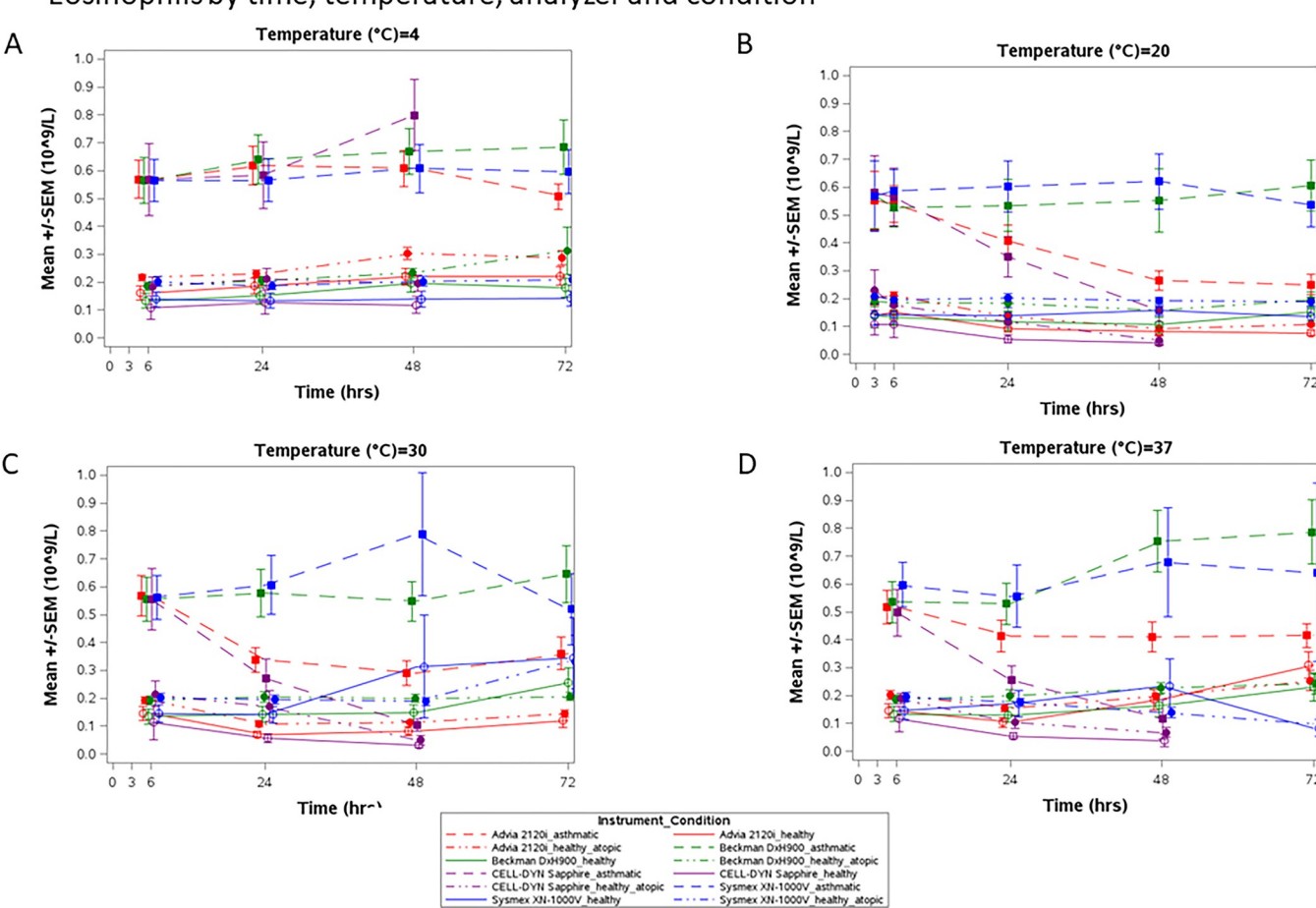

**Fig 5. Comparison of absolute eosinophil count for those with eosinophilic asthma (dashed line), healthy (with atopy (dashed/dotted line) and without atopy (solid line) participants by the 4 hematology analyzers by time and temperatures.** A. Samples stored at 4°C, B. stored at 20°C, C. stored at 30°C and D. stored at 37°C. The baseline is shown in the 20°C (B) plot and mean eosinophil counts for those with eosinophilic asthma ranged between 0.552–0.579 cells x10$^9$ /L, healthy participants with atopy at 0.190–0.232 cells x10$^9$/L, and healthy without atopy at 0.106–0.142 cells x10$^9$/L, depending on analyzer (S6 Table). Larger absolute drops are observed in the Sapphire and 2120i eosinophil counts for participants with eosinophilic asthma after 24h at 20°C and 30°C for both analyzers and also at 37°C for the Sapphire; moving them into the healthy range. More extensive variability (outliers) is observed at higher temperatures for the XN-1000V after 48h. No platform maintains reliability to measure eosinophil count across all storage times and temperatures.

the predicted eosinophil levels were higher than expected in samples with lower baseline levels than expected for these two analyzers.

### Flow cytometry analysis of eosinophils

Activation, maturation, and live-dead markers on the eosinophils, measured by flow cytometry, are shown in Fig 6, with summary statistics in S9 Table and differences in LS means in S10 Table. At 4°C, live-dead aqua, CD62L (l-selectin), CD123 and FcεR1 remained stable over time and CD66b and CD11b (the maturation markers) increased with time by 24h. At 20°C, live-dead aqua and FcεR1 remained stable; CD11b and CD66b increased with time but less than at 4°C. CD123 also increased with time and CD62L (l-selectin) decreased over time by 24h. At 30°C, live-dead aqua, FcεR1 and CD123 were stable until 72h when levels increased dramatically, CD62L (l-selectin) decreased over time more dramatically than at 20°C and CD11b and CD66b also decreased over time at this temperature.

FACS data LS means

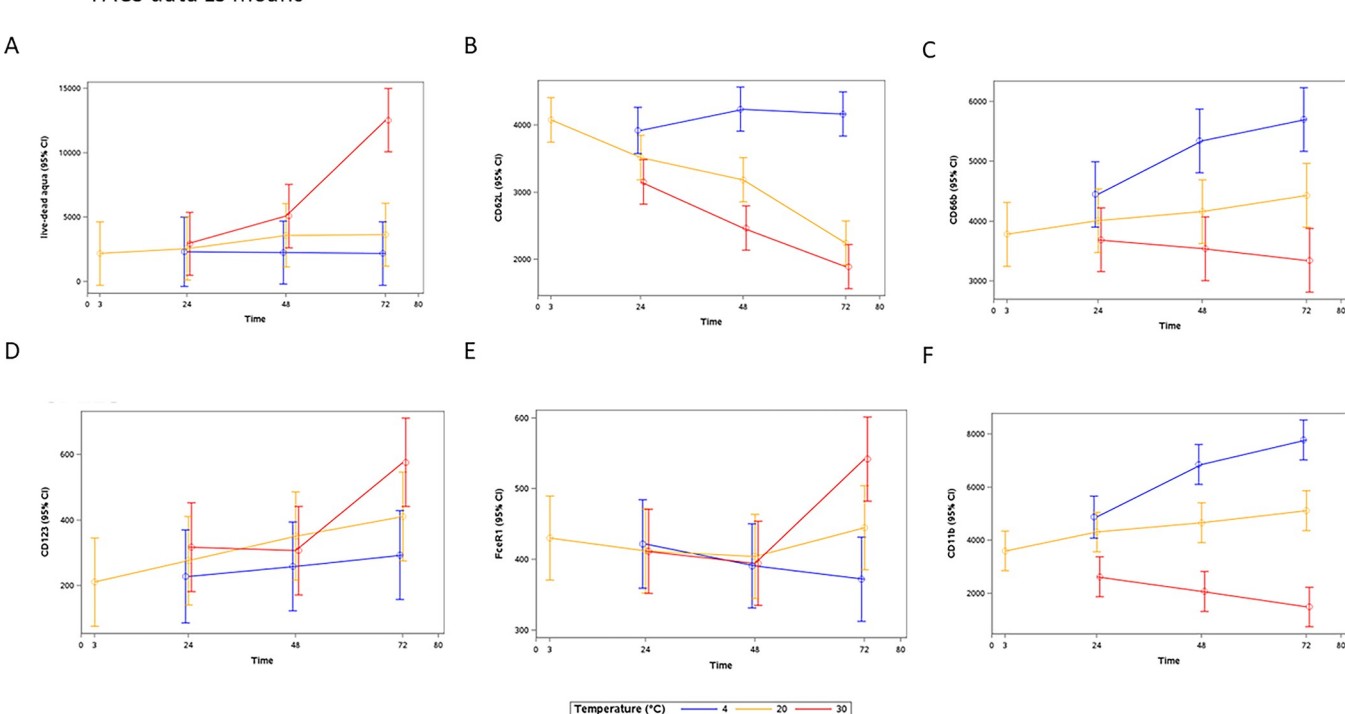

**Fig 6.** Activation/maturation markers measured on eosinophils using flow cytometry show activation and death with time and temperature (LS Means±95% CI, n = 12 (6 with eosinophilic asthma, 3+3 healthy with and without atopy)) A. live/dead aqua staining, B. CD62L (l-selectin), C. CD66b, D. CD123, E. FcεR1, and F. CD11b. Results of the linear mixed models (differences in LS means *versus* 3-hour and 20˚C baseline) for each marker by time point and temperature are presented in S10 Table. Live-dead aqua, CD123 and FcεR1 increased dramatically at 30˚C by 72h. CD66b and CD11b increased at 4˚C and decreased at 30˚C over time. CD62L decreased over time at both 20˚C and 30˚C.

## Indication of activation by granule release from eosinophils

Eosinophil-derived neurotoxin (EDN), a commonly used biomarker of eosinophil activation, was relatively stable in samples kept at 20˚C up to 72h post sampling (Fig 7, S10 Table), but found to be increased in plasma from 48 hours post-sampling for samples kept at higher temperatures. The difference between mean at baseline (3h, 20˚C) and 48h at 37˚C was -224ng/mL (95%CI -322, -125ng/mL). By 72 hours at 30˚C it was -132ng/mL (-231, -34ng/mL) and at 37˚C, -478ng/mL (-579, -376ng/mL). The non-statistically significant increase in EDN release at 4˚C coincided with the activation markers CD11b and CD66b (Fig 6). The higher EDN release observed in the eosinophilic asthma patients upon stimulation was lost *versus* healthy participants when normalized to eosinophil baseline count (Fig 7B), indicating similar EDN levels per eosinophil independent of disease/activation status. S12 Fig demonstrates the better correlation for serum EDN with baseline eosinophil count (Spearman r = 0.7800) *versus* plasma EDN values (r = 0.5239). However, the extremes (the participants with the lowest and highest eosinophil counts) drive the poorer correlation for plasma EDN. There was no difference in EDN release between shaken and non-shaken samples (S13 Fig).

## Discussion

This is the first direct head-to-head comparative study of white cell differential measurements using Abbott, Beckman, Siemens, and Sysmex hematology analyzers. Our results show that the four hematology platforms report similar white cell differential counts if the blood sample is

EDN data

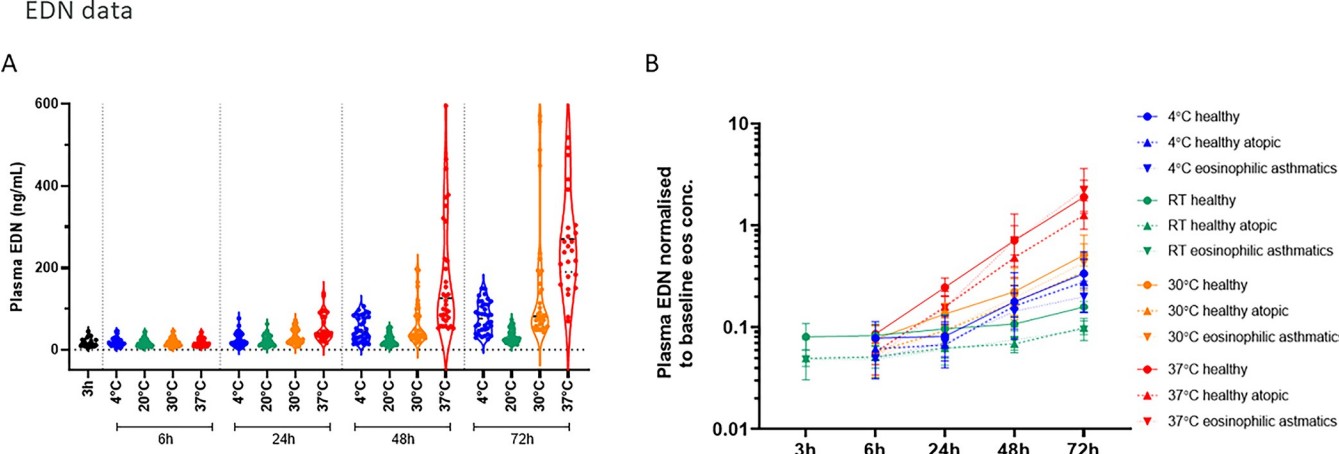

**Fig 7. Eosinophil-derived neurotoxin (EDN) in plasma.** A. EDN release with time and temperature, data merged from healthy (n = 12) and asthmatic patients (n = 6), shaken, and non-shaken samples. B. EDN normalized to eosinophil count per donor at baseline. Each data point represents geometric mean ±SD, n = 12 per time point 6-72h (shaken and non-shaken), n = 4–6 at 3h.

analyzed within the day of collection. However, we found significant differences in their ability to report stable eosinophil, basophil, and neutrophil counts over time, starting already the day after blood sampling. We have also demonstrated that the temperature at which the samples are stored can exaggerate these temporal differences. Health status (healthy versus atopy/ asthma) and mechanical stress (mimicking real-world transportation), however, did not affect the cell count results. This study highlights the importance of understanding the performance of the selected platform and cell type of interest to select the optimal analyzer, time to analysis and sample handling conditions for adequate results in clinical practice. It further emphasizes the importance of routinely measuring whole blood differentials on the day of sampling, as is the typical performance in clinical practice.

In short, this study demonstrates that the Abbott CELL-DYN Sapphire performs well on refrigerated samples for all cell types up to 48 hours post-collection. However, our results suggest that it loses its ability to identify and count eosinophils, monocytes, and platelets by 24–48 hours in samples stored at temperatures of 20˚C and higher. The Siemens ADVIA 2120i is sensitive to sample storage temperatures for all cell types with time (neutrophils in refrigerated samples and eosinophils, lymphocytes, monocytes, and basophils at temperatures from 20˚C and higher), except for platelets which are adequately counted at 4–30˚C up to 72 hours post sampling. The Sysmex XN-1000V and Beckman Coulter DxH900 analyzers behave quite similarly for all cell types. Still, their cell count results vary by time for neutrophils at low and high temperatures and for lymphocytes and monocytes at high temperatures whereas platelets are adequately assessed. All four devices showed variance over time for cell differentials in blood stored at 37˚C, especially at 48-72h, which has been reported to be a consequence of cell degeneration and death [12, 13].

Leukocytes are sensitive to external temperatures with time [14], which causes changes in their commonly used identification markers and morphology; therefore, it may be a challenge for existing technology (light scatter, coloring, and gating together with the selection of markers) to identify these cells correctly in aged samples. All hematology platforms in the present study are affected by this biological issue, though with different results depending on the technology used. This investigation concludes that each technological platform has different sensitivities to cell changes, which are highly influenced by time from sampling to analysis and temperature. Although we don't propose a preferred hematology platform, awareness of the

external factors which might influence the analysis is essential to improving control and minimizing the divergence from a freshly analyzed blood sample [6, 7].

Focusing on the eosinophils, given its importance in asthma management, there are significant differences in the method by which the four technology platforms recognize and count these cells, which corresponds to the eosinophil activation status and morphology (e.g., size and granule content). This is likely the main reason for the difference observed between the devices. There was no difference in counts between samples exposed to shaking *versus* those not shaken, in any of the platforms, suggesting that eosinophils may not be highly sensitive to mechanical stress caused by transportation in the real-world context. With regards to the health status (healthy w/wo atopy and eosinophilic asthma patients), the hematologic platforms delivered similar eosinophil counts, indicating that health status/activation does not affect the analyzer's ability to identify and count the cells. The exception was the ADVIA 2120i at 4°C, where an overestimated eosinophil number was observed for those with asthma. We found that this was due to neutrophil infiltration into the eosinophil gate.

Overall, the Sysmex XN-1000V and Beckman Coulter DxH900 technology platforms performed best in terms of reporting the most stable eosinophil counts over time and at different temperatures. The XN-1000V platform was superior with regards to stability because we saw only random outliers at higher temperatures (30 and 37°C) whereas the DxH900 showed on average stable measurements, but eosinophil numbers tended to increase with time (statistically significant differences, however not likely clinically meaningful). It also showed more extensive interindividual variability which was especially prominent at 20°C (giving non-statistically significant changes from the 3h baseline for this temperature). The CELL-DYN Sapphire platform was, as expected [4, 14], the most sensitive hematology analyzer to time and temperature, and quickly (within 24h) lost its ability to stably count eosinophils if blood samples were kept at room temperature (20°C) or above. In contrast, refrigerated samples were stably measured up to at least 48 hours. Longer stability measures are warranted to give the upper time limit for reliable cell differentials by the CELL-DYN Sapphire analyzer. The 2120i reported dramatically higher eosinophil counts in refrigerated samples by as much as 560 cells/μL in some individuals and 286 cells/μL on average. The scatter images identified the root cause where the distinct baseline neutrophil population collapses due to altered morphology by apoptosis and infiltrates the eosinophil gate. At 20°C and 30°C, the 2120i instead reported lower eosinophil counts already within 24h by on average as much as 31% (48% at 48h), similar to the Sapphire platform (38% and 60% at 24 and 48h, respectively). This apparent loss of ability by the Sapphire and 2120i analyzers to identify and count eosinophils in blood kept at 20°C and higher was apparent independent of baseline eosinophil level, albeit the absolute reduction was more prominent in those asthma patients with the highest levels of eosinophils which may impact more in clinical practice in the identification of potential responders to eosinophil-targeting therapies.

In the evaluation of eosinophil activation, maturation, and death processes by flow cytometry, we were able to identify increased death signaling already 24 hours post sampling at 20°C and above (however, with the variability/spread in data it was not statistically significant until 72h at 37°C). This was mirrored in counting live cells where eosinophils decreased in numbers from 24 hours at 20°C. The levels of activation markers were found to be higher in the samples from the eosinophilic asthma patients (n = 6) compared to the healthy volunteers (n = 3+3) but were due to the higher number of eosinophils. The activation/maturation markers CD11b, CD66b, CD62L, and CD123 increased in line with eosinophil death (live-dead aqua staining). We further observed increased EDN release not only at the higher temperatures (30°C and 37°C) with time but also at 4°C, which followed the same pattern as the increase in CD11b and

CD66b, indicating activation of eosinophils still occurs with refrigeration. Despite this, their ability to be counted could be maintained (e.g., by the Sapphire platform).

As to the differences in performance between the analyzers, all 4 hematology platforms use flow cytometry with laser light sources and detection of scattered light. The similarities and differences between the platforms in how they identify and count the different cell types depend on the details of the technology and cell identifiers used and/or algorithms for data analysis, however the specific details are not publicly available. For example, the similar assessment of eosinophils by the Sapphire and 2120i *versus* XN-1000V and DxH900 indicates differences in specific eosinophil and/or neutrophil characteristics as demonstrated by the observed increase in blood eosinophil count for healthy participants when stored at 4˚C. Moreover, platelets are counted differently by platforms that use impedance (Sysmex, Beckman) *versus* an optical method (Siemens) or the one using both (Abbott), thus indicating that background technology matters, however, differently for different cell types.

Mechanical stress did not alter the release of EDN, as with the cell count, which was mainly time and temperature dependent. We observed that the EDN content in plasma (and serum) correlated well to the eosinophil cell count, indicating a similar capacity per cell to release EDN independent of disease status. As expected, EDN is released upon activation and induced cell death, thus commonly used as an activation marker; and this release is similar in healthy volunteers and patients with eosinophilic asthma. We also observed a better correlation between EDN content in serum to eosinophil count *versus* what was seen in the plasma samples, indicating that serum may be the better, less variable matrix for EDN determination.

Our study has several limitations; firstly, the mechanical stress of vortex shaking was a superficial, standardized way to explore the impact of rough handling, like in the real-world transportation of samples. However, in the real-world context, this is impossible to control. Hence, our data only show that agitation twice a day does not affect the ability of the analyzers to identify and count cell differentials. Secondly, the CELL-DYN Sapphire results are from separate test tubes *versus* the rest of the platforms and analyzed in another lab. The samples were taken and analyzed at the same time and kept under the same conditions but were handled by a different scientist; however, the CELL-DYN Sapphire eosinophil counts *versus* the ADVIA and Sysmex platforms are consistent with previously published results [4, 14], which strengthens our results as now produced by independent labs. Thirdly, for practical reasons (maximal number of samples to be handled within preset time restrictions of 10% deviation and logistical challenges), the study was conducted in 3 separate experiments with n = 6 within a 2-month time window and the maximum number of donors was thus limited to a total of 18 (6+6+6). The first 2 experiments were done on healthy volunteers w/wo atopy (n = 3+3 per experiment) whereas the final experiment was performed on blood from patients with asthma (n = 6). Lastly, the flow cytometry analysis was not performed on shaken samples or those kept at 37˚C as in a previous study we saw that most cells are dead or dying at 37˚C *ex vivo*. Capturing the activation and death status at 30˚C was therefore prioritized.

To our knowledge, this is a unique study investigating blood cell differentials in four different hematology analyzers head-to-head at multiple conditions and health statuses. The data should be used to optimize future performance, and help building strategies for a better recognition of blood cell count in clinical practice and trials.

## Conclusions

The lessons learnt from this study should be implemented to optimize the handling conditions of fresh blood samples when there will be a delay between collection and testing for cell count. We recommend that whenever possible, whole blood cell differentials are analyzed within the

same day of sampling to avoid variability in results caused by preanalytical factors in combination with the hematology analyzer used. We further recommend careful consideration of the shipment temperature and time until analysis, in combination with the choice of hematology platform to be used and the main cell of interest, to optimize performance and data quality for when analysis within the day of sampling cannot be performed.

## Supporting information

**S1 Table. Participant demographics.**
(PDF)

**S2 Table. FACS panel and eosinophil count in the FACS analysis (% eos/total white blood cells, Mean±SD).**
(PDF)

**S3 Table. Model results for all baseline (3h) cells counts ($x10^9$ cells/L) per technology platform.**
(PDF)

**S4 Table. Differences ($x10^9$ cells/L) from baseline (3h, 20˚C) within analyzer for all cell types.**
(PDF)

**S5 Table. Variance estimates from MMRM analyses.**
(PDF)

**S6 Table. Eosinophil count ($x10^9$ cells/L) at 3 hours and 20˚C by condition and analyzer.**
(PDF)

**S7 Table. Eosinophils–change from baseline at 24 and 48 hours (20˚C).**
(PDF)

**S8 Table. Eosinophils–percentage change from baseline at 24 and 48 hours (20˚C).**
(PDF)

**S9 Table. Summary statistics for absolute FACS markers and EDN by time and temperature.**
(PDF)

**S10 Table. Differences from baseline (3h, 20˚C) for FACS markers and EDN.**
(PDF)

**S1 Fig. Correlations between 3h (baseline, 20˚C) and 6h (all temperatures) cell differentials with the 4 different analyzers.**
(PDF)

**S2 Fig. WBC variability with time, temperature and mechanical stress (shaken) in the 4 different platforms.**
(PDF)

**S3 Fig. LS means–platelets by time, temperature and instrument (excluding Beckman and Sysmex results for subject 10 at 4˚C, 24 hours, shake = no).**
(PDF)

**S4 Fig. LS means–neutrophils by time, temperature and instrument.**
(PDF)

**S5 Fig. LS means–lymphocytes by time, temperature and instrument.**
(PDF)

**S6 Fig. Eosinophils by time, temperature, analyzer and condition.**
(PDF)

**S7 Fig. Eosinophil count by the Abbott CELL-DYN Sapphire and Siemens ADVIA 2120i is independent of baseline levels (asthmatic participants only).**
(PDF)

**S8 Fig. Predicted eosinophils (best linear unbiased prediction) by baseline eosinophils and analyzer.**
(PDF)

**S9 Fig. Representative scattergram from the 2120i for 4 C˚- & 20C˚-stored asthmatic blood sample.**
(PDF)

**S10 Fig. LS means–monocytes by time, temperature and instrument.**
(PDF)

**S11 Fig. LS means–basophils by time, temperature and instrument.**
(PDF)

**S12 Fig. Serum vs plasma EDN–correlation with Eos count at baseline.**
(PDF)

**S13 Fig. EDN release in shaken versus non-shaken samples.**
(PDF)

**S14 Fig. FACS gating strategies.**
(PDF)

## Acknowledgments

We are incredibly grateful to Helena Orre Ekdahl and Ruth Wickelgren at Clinical Chemistry, Sahlgrenska University Hospital, Gothenburg, Sweden, for the analysis of cell differentials in the CELL-DYN Sapphire platform; Mireille Backers & Cecilia Söderberg at Beckman Coulter for letting us borrow a DxH900 machine over the course of the study; Madeleine Rådinger and Helén Törnqvist at Krefting Medical Centre and Katarina Vesterlund for great collaboration and assistance with the collection of blood samples from people with eosinophilic asthma and healthy volunteers, respectively; Sofia Lundin, Karin Gedda, Ida Gidlöf, Gabriela Baeza, Josephina Edsbagge and Kinga Balogh Sivars for Ethical approvals and contracts. Manuel Perera Chang, CLINCHOICE INC, for scientific input. Marthe Wedø Aune, St Olavs Hospital, for sharing experience and results from comparing >80 blood samples in 3 of our investigated platforms at 2 different temperatures. Finally, Lee Wulund at AstraZeneca for providing expert manuscript and publication advice on this manuscript.

## Author Contributions

**Conceptualization:** Annika Åstrand, Johan Mattsson, Lina Odqvist, Steven Eck, Glen Hughes, Gabriela Luporini Saraiva, Anna Schantz, Ioannis Psallidas, Christopher McCrae.

**Data curation:** Annika Åstrand, Cecilia Wingren, Claire Walton, Johan Mattsson, Komal Agrawal, Madelene Lindqvist, Björn Burmeister.

**Formal analysis:** Annika Åstrand, Claire Walton, Johan Mattsson, Komal Agrawal, Madelene Lindqvist.

**Funding acquisition:** Annika Åstrand.

**Investigation:** Annika Åstrand, Cecilia Wingren.

**Methodology:** Annika Åstrand, Cecilia Wingren, Claire Walton, Johan Mattsson, Komal Agrawal, Madelene Lindqvist, Björn Burmeister.

**Project administration:** Annika Åstrand, Cecilia Wingren.

**Resources:** Annika Åstrand, Cecilia Wingren, Lina Odqvist, Christopher McCrae.

**Supervision:** Annika Åstrand, Cecilia Wingren, Steven Eck, Glen Hughes, Anna Schantz, Christopher McCrae.

**Validation:** Annika Åstrand, Cecilia Wingren, Claire Walton.

**Visualization:** Annika Åstrand, Johan Mattsson.

**Writing – original draft:** Annika Åstrand, Claire Walton, Johan Mattsson, Anna Schantz.

**Writing – review & editing:** Annika Åstrand, Cecilia Wingren, Claire Walton, Johan Mattsson, Komal Agrawal, Madelene Lindqvist, Lina Odqvist, Björn Burmeister, Steven Eck, Glen Hughes, Gabriela Luporini Saraiva, Anna Schantz, Ioannis Psallidas, Christopher McCrae.

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
