## [Decision Letter · Decision Letter 0]

18 Dec 2023

PONE-D-23-30649A comparative study of blood cell count in four automated hematology analyzers: an evaluation of the impact of preanalytical factorsPLOS ONE

Dear Dr. Åstrand,

Thank you for submitting your manuscript to PLOS ONE. After careful consideration, we feel that it has merit but does not fully meet PLOS ONE’s publication criteria as it currently stands. Therefore, we invite you to submit a revised version of the manuscript that addresses the points raised during the review process.

We look forward to receiving your revised manuscript.

Kind regards,

Enoch Aninagyei, PhD

Academic Editor

PLOS ONE

“The authors, study participants and collaborators have no competing interest in the results. AA, CeW, ClW, JM, KA, ML, LO, BB, SE, GH, GLS, AS, IP, and CMcC are or were employees of AstraZeneca at the time of study conduct and may own stock/stock options.”

Reviewers' comments:

Reviewer's Responses to Questions

**Comments to the Author**

1. Is the manuscript technically sound, and do the data support the conclusions?

Reviewer #1: Yes

Reviewer #2: Yes

2. Has the statistical analysis been performed appropriately and rigorously? 

Reviewer #1: I Don't Know

Reviewer #2: Yes

3. Have the authors made all data underlying the findings in their manuscript fully available?

Reviewer #1: Yes

Reviewer #2: Yes

4. Is the manuscript presented in an intelligible fashion and written in standard English?

Reviewer #1: Yes

Reviewer #2: Yes

5. Review Comments to the Author

Reviewer #1: The authors compare 4 hematology Analyzers in whole cell blood count, but most of clinical laboratory has one or two analyzer that they are working.

The authors neither experiment nor discuss about RBC, HCt, Hgb and RBC index's. may be they will do it in their next manuscript.

the aim of this study is not clear.

Reviewer #2: Well written and well formatted article. Technically sound.Excellent work.

The methodology is sound,However the sample size could have been higher to establish more conclusive findings.This helps to identify possible pre analytical variables and their influence on outcomes

6. PLOS authors have the option to publish the peer review history of their article (what does this mean?). If published, this will include your full peer review and any attached files.

Reviewer #1: No

Reviewer #2: **Yes: **Bobby Abraham

---

## [Author Response · Author response to Decision Letter 0]

22 Dec 2023

Response to the Reviewers:

Reviewer #1: The authors compare 4 hematology Analyzers in whole cell blood count, but most of clinical laboratory has one or two analyzer that they are working.

The authors neither experiment nor discuss about RBC, HCt, Hgb and RBC index's. may be they will do it in their next manuscript.

the aim of this study is not clear.

Response: We focused our study on the White Blood Cells as stated in the beginning of the paper, and albeit more information is achieved in the analysis for all hematology platforms, we did not summarize the information for RBCs, HCt, Hgb etc. This could be done in a separate project by using the existing data.

We have clarified (Abstract and Introduction) 1) the gap in knowledge regarding aging blood samples and pre-analytical factors in the public domain and 2) the aim of our study to provide solid data that would help with future cell differential evaluations.

Reviewer #2: Well written and well formatted article. Technically sound.Excellent work.

The methodology is sound,However the sample size could have been higher to establish more conclusive findings.This helps to identify possible pre analytical variables and their influence on outcomes

Response: We agree, it would have been preferred to have a larger number of donors in our study, but for technical limitations (the ability to analyse samples within a certain time frame), we had to limit the study to n=6 per experiment. We had previous knowledge internally regarding variability in the measurements and also whether there seemed to be differences between high/low eos subjects, in disease and in healthy individuals. Our power calculations, taking into account the repeated measures per sample and the multiple conditions, indicated that an n=18 would be adequate to compare between platforms for solid results and the n=6 per subject condition would be enough to identify any major changes similar to the size of differences found between platforms. As seen in the results, the subject condition does not affect the ability of the platforms to identify and count white blood cell differentials, nor did the mechanical stress make any difference, so these results have been averaged. Taken together, each donor has very strong data and this is included and explained in the analysis section.

We have extended the section on ‘Limitations of the study’ to highlight that with the identification of no interference of disease status or effect of mechanical damage, and that sample size was extensively increased by merging results for the different conditions between platforms.

---

## [Decision Letter · Decision Letter 1]

29 Jan 2024

PONE-D-23-30649R1A comparative study of blood cell count in four automated hematology analyzers: an evaluation of the impact of preanalytical factorsPLOS ONE

Dear Dr. Åstrand,

Thank you for submitting your manuscript to PLOS ONE. After careful consideration, we feel that it has merit but does not fully meet PLOS ONE’s publication criteria as it currently stands. Therefore, we invite you to submit a revised version of the manuscript that addresses the points raised during the review process.

We look forward to receiving your revised manuscript.

Kind regards,

Enoch Aninagyei, PhD

Academic Editor

PLOS ONE

Reviewers' comments:

Reviewer's Responses to Questions

**Comments to the Author**

1. If the authors have adequately addressed your comments raised in a previous round of review and you feel that this manuscript is now acceptable for publication, you may indicate that here to bypass the “Comments to the Author” section, enter your conflict of interest statement in the “Confidential to Editor” section, and submit your "Accept" recommendation.

Reviewer #3: (No Response)

2. Is the manuscript technically sound, and do the data support the conclusions?

Reviewer #3: Partly

3. Has the statistical analysis been performed appropriately and rigorously? 

Reviewer #3: Yes

4. Have the authors made all data underlying the findings in their manuscript fully available?

Reviewer #3: No

5. Is the manuscript presented in an intelligible fashion and written in standard English?

Reviewer #3: No

6. Review Comments to the Author

Reviewer #3: The authors have compared the counts of white blood cells in different hematology platforms in terms of:

1- Storage temperature

2- Time to analysis

3- Mechanical stress

With later a special focus on eosinophils.

This comparative study is needed to provide insight into the factors that contribute to the variability of results across platforms.

There are a number of concerns, outlined below, that need to be addressed in order to publish this to be of use to the scientific and medical community.

1- The authors need to justify the focus on eosinophils. Perhaps stating in the introduction.

2- The experimental design is a little confusing and hard to follow. I suggest adding a flow chart of the different participants and their characteristics and different experiments and manipulations done.

3- Please define what you mean by baseline in the methods section.

4- In line 109 it is mentioned that two tubes from each donor were put in 4 different temperature. Were each tube split into two different tubes in order to have them in 4 different temperatures? This is not clear. There are many instances where the methodology of handling the samples is not clear. Therefore, it will be hard for other scientists to reproduce the protocol used.

5- In line 122, RT, which I assume is refereeing to room temperature, was not defined prior to this point

6- I have a concern regarding patch effects of the different experiments run at different places and different times and yet they were compared to each other in the results. Is there a way statistically to control for that?

7- In Line 133, “(w/wo)” means with or without atopy? Is there a reason for not using one, either with or without, or using both groups separately; so the healthy will have n=6 (3+3).

8- For the flow cytometry and EDN assay, were there technical replicates? If not please justify.

9- In line 167, it is indicated that the diluted samples had an upper range of 640 ng/ml which is outside the standard curve. It is not very clear this section in the methods as there are many different brackets and they are not always properly used and effects the understanding the sentences. This is seen all-over the manuscript and needs revisions.

10- The results section is disorganized and very hard to follow. it needs to be organized with clear subheadings and better flow. For example, in the first section of the results, after the demographics of the participants, is not very clear what experiments our question is being answered especially that it feels like some the results are just repeated in other sections. Moreover, the description of the results between the different platforms and the different temperature and times need to be written in a more clear. For example, when the increase is mentioned by X (y-z) are you referring to a percent increase and a range. This need to be added in the statistical analysis section, where you mention that data are presented as ………. The results section needs major reorganization to help the readers follow the logic of the experiments.

11- The discussion need to describe the results in more details. For example, in line 358, the authors need to rewrite this to better descibe their findings in the different experiments. Also, in line 366, the authors state that leukocytes are sensitive to external temperatures; was this based on the their data or from the literature? They need to specify that.

12- Discuss the significance of examining eosinophils specifically.

7. PLOS authors have the option to publish the peer review history of their article (what does this mean?). If published, this will include your full peer review and any attached files.

Reviewer #3: **Yes: **Sameera Abuaish

---

## [Author Response · Author response to Decision Letter 1]

13 Mar 2024

Response to Reviewer 3 is given in the new version - many thanks for thorough review and constructive feedback

---

## [Editor Report · Decision Letter 2]

25 Mar 2024

A comparative study of blood cell count in four automated hematology analyzers: an evaluation of the impact of preanalytical factors

PONE-D-23-30649R2

Dear Dr. Annika Åstrand,

We’re pleased to inform you that your manuscript has been judged scientifically suitable for publication and will be formally accepted for publication once it meets all outstanding technical requirements.

Kind regards,

Enoch Aninagyei, PhD

Academic Editor

PLOS ONE
---

## [Editor Report · Acceptance letter]

8 Apr 2024

PONE-D-23-30649R2 

PLOS ONE

Dear Dr. Åstrand, 

I'm pleased to inform you that your manuscript has been deemed suitable for publication in PLOS ONE. Congratulations! Your manuscript is now being handed over to our production team.

Kind regards, 

on behalf of

Dr Enoch Aninagyei 

Academic Editor

PLOS ONE